# A Case of Psoriatic Disease and Hidradenitis Suppurativa in a Child with Chromosome 17q21.31 Microduplication Syndrome

**DOI:** 10.3390/children10060931

**Published:** 2023-05-25

**Authors:** Ersilia Tolino, Nevena Skroza, Emanuela Del Giudice, Patrizia Maddalena, Nicoletta Bernardini, Ilaria Proietti, Alessandra Mambrin, Federica Marraffa, Giovanni Rossi, Riccardo Lubrano, Concetta Potenza

**Affiliations:** 1Department of Medico-Surgical Sciences and Biotechnologies, Dermatology Unit “D. Innocenzi”, Sapienza University of Rome, Polo Pontino, Via Firenze, 1, 04019 Terracina, Italy; 2Maternal and Child Health Department, Santa Maria Goretti Hospital, Sapienza University of Rome, Polo Pontino, 04100 Latina, Italy

**Keywords:** psoriatic arthritis, hidradenitis suppurativa, 17q21.31 microduplication syndrome ù

## Abstract

Psoriatic disease is a chronic, relapsing inflammatory disorder, characterized mostly by cutaneous erythematous scaly plaques sometimes associated with arthritis. Hidradenitis suppurativa (HS) is a chronic relapsing inflammatory disease of the apocrine glands, characterized clinically by painful abscesses, sinus tracts and scars. It typically occurs after puberty, affecting mainly intertriginous areas of the body. There is a strong association between HS and psoriasis since they share the same pathogenic inflammatory pathway. The patient presented: low birthweight, microcephaly, facial dysmorphisms, lumbar hyperlordosis, walking difficulties, global psychomotor developmental delay and learning disabilities. A genetic evaluation revealed a 2.5 Mb de novo microduplication in the 17q21.31 chromosomal region. Dermatological examination revealed HS (Hurley stage II-HS) distributed in the genital area and inguinal folds, psoriatic plaques on the retroauricolar folds, on the elbows bilaterally and on the lateral aspect of the right ankle and psoriatic arthritis. The patient was treated with adalimumab, with a marked improvement of both conditions. To our best knowledge, we report the first case of coexisting Psoriatic Arthritis Disease and Hidradenitis Suppurativa in a child with chromosome 17q21.31 microduplication syndrome. We hypothesize that gene CRHR1 duplication included in the 17q21.31 chromosomal region might be involved in the pathogenesis of both diseases.

## 1. Introduction

Psoriatic disease is a chronic relapsing inflammatory disorder, characterized mostly by cutaneous erythematous scaly plaques sometimes associated with arthritis and an increased rate of cardiometabolic, hepatic and psychological comorbidity [1]. Topical therapy is the mainstay treatment option with limited adverse effects. Recently, biologics have renewed the conventional psoriasis treatment, achieving a greater efficacy and increased patient acceptance due to the decreased adverse event [2]. Epidemiology of pediatric psoriasis is uncertain and large studies are lacking. In Europe, a prevalence rate ranging between 0.17 and 1.5% has been reported [3]. Clinical presentation of pediatric psoriasis seems to be quite different from adults, especially in infants aging less than 1 year. In such cases, the most affected areas are diaper areas and inguinal folds (psoriatic diaper rash). Single lesions are usually smaller, less infiltrating, and less desquamated than those observed in adults. As they age, clinical manifestations resemble those of adults, with the most common areas affected (scalp, face, elbows, and knees) with “classic” erythematous and scaly plaques. Similarly to the adults form, psoriasis in children is associated with a series of metabolic, autoimmune and psychiatric comorbidities (obesity, diabetes, hypertension, arthritis, depression and anxiety) [4]. Moderate-to-severe pediatric psoriasis represents a challenge regarding its treatment. Conventional systemic drugs (acitretin, cyclosporin, and methotrexate) have potential severe side effects, such hematological, renal and hepatic toxicity, and teratogenicity. In addition, they require frequent laboratory monitoring. Thus, biologics represent a promising treatment option even for pediatric patients. Adalimumab, etanercept, ixekizumab, secukinumab, and ustekinumab have been approved for children, showing high efficacy and a favorable side effect profile [5].

Hidradenitis suppurativa (HS) is a chronic relapsing inflammatory disease of apocrine glands, characterized clinically by painful abscesses, sinus tracts and scars. It typically occurs after puberty, affecting mainly intertriginous areas of the body [6]. Observed comorbidities fall into several categories: cardiovascular diseases, inflammatory and autoimmune diseases, hormone-related disorders and psychiatric illnesses [7]. HS results in significant quality of life impairment, and is more profound when comparing most other chronic skin diseases. Pain, diagnostic delay, comorbidities, chronicity and other HS symptoms have the greatest influence on health-related quality of life [8]. Topical and systemic antibiotics, acitretin, disruptive therapies (surgery, laser) and light therapy have been used for HS with variable results. There is no surgical optimal therapy; however, several options are available with an individualized approach depending upon various factors, such as the chronicity and extent of disease, affected site, presence of long-standing lesions, and patient comorbidities [9].

Surgical treatment in HS ranges from procedural treatments (e.g., laser) and minor surgery (e.g., incision, drainage and deroofing) to major surgery (e.g., wide local excision) [10]. Very recent methods include reconstructive surgery through a co-graft of Acellular dermal matrix and split thickness skin graft for a better wound closure after a wide excision [11,12].

Adalimumab (a fully human, high-affinity, recombinant anti-tumor necrosis factor (TNF) alpha-monoclonal antibody) is the only US Food and Drug Administration (FDA) approved biologic agent for moderate-to-severe HS, but new therapeutic options are being studied that target different, specific cytokines that are involved in HS pathogenesis (i.e., IL-17) [13]. 

The efficacy of Adalimumab goes beyond the reduction of the disease severity score scores like PASI for psoriasis and Hurley stage for HS. Regarding psoriasis, Adalimumab revealed efficacious in improving dermatology-specific and general health-related quality of life, work and activity limitations, and psoriasis-related symptoms. 

Besides, real-life studies showed a marked improvement in terms of mental and physical well-being, cutaneous body image, anxiety, depression and psoriasis severity, which were maintained at 6-months [14]. 

Similar findings were reported in HS patients under Adalimumab. In a real-life setting, patients reported decreased skin pain and improvements in both physical and psychological aspects of QoL and an improvement in work productivity [15]. Assessment of therapeutic responses more difficult because some lesions (especially fistulas) are not clinically evaluable. For this reason, ultrasonography has emerged as an useful tool to evaluate treatment response with multiple parameters (degree of oedema, fibrosis and the distribution of vascularization assessed by eco-color-Doppler). Furthermore, ultrasonography may have a role in leading therapeutic decisions on the base of the activity signal of HS lesions [16]. 

Adalimumab has showed to decrease C-reactive protein levels in psoriasis and HS; this reduction is associated with percentage reductions in PASI. This can have an impact on psoriasis comorbidities. In fact, psoriatic arthritis, such as obesity, diabetes mellitus, dyslipidemia, arterial hypertension, inflammatory bowel disease, and depression are all characterized by a low-grade inflammation, whose best biomarker is C-reactive protein. In general, C-reactive protein could be a valid biomarker of response to adalimumab both for psoriasis and hidradenitis [17,18,19]. 

Pediatric HS is rare. In particular, less than 2% of patients with HS have disease onset before age 11 [20]. Pediatric patients with HS face several comorbidities. These include metabolic and endocrine abnormalities (obesity and polycystic ovary syndrome), acne vulgaris, anxiety or depression disorder, asthma, and other reactive airway diseases [21]. Treatment in pediatric patients aims to alleviate pain, minimize inflammation and scarring, prevent disease progression, and postpone the need for surgery. Topical and oral antibiotics can be used for mild and moderate cases, whereas severe forms can be treated with biologic agents, laser therapy or surgery [22].

There is a strong association between HS and psoriasis since they share the same pathogenic inflammatory pathway: upregulation of cytokines TNFa, IL-12/23 and IL-17. This implies that they could respond both to the same biologic treatments [21]. Pediatric cases of HS are often elicited by endocrine diseases, such as congenital adrenal hyperplasia, premature adrenarche or androgen secreting tumors [23]. HS can also be associated with autoinflammatory syndromes: pyoderma gangrenosum, PASH, PAPASH and PsAPASH [18]. Furthermore, it has been demonstrated that there is an association between HS and several genetic disorders: Down syndrome, keratitis-ichthyosis-deafness syndrome (KID syndrome), Dowling-Degos disease, Basex-Duprè-Christol syndrome, Smith-Magenis syndrome, Familial Mediterranean Fever, Myotonic Dystrophy and trisomy 1q [24,25,26].

Beyond these syndromic disorders, familial forms of HS have been reported. These cases are characterized by a strong family history and established underlying monogenic etiology. The implicated genes driving familial HS are mainly members of the γ-Secretase Complex Protein-Coding Genes. This complex is essential in the activation of Notch signalling pathways, and is generated from four subunit domains: Nicastrin (NCSTN), Presenilin Enhancer 2 (PEN2), Presenilin 1 (PSEN1) or PSEN2 and Anterior Pharynx Defective (APH) 1A or [27].

Many studies have reported an association between psoriasis and hidradenitis suppurativa. However, some evidence seems to be controversial [28]. In patients with a coprevalence of HS and Pso, there is an increased risk in the development of obesity and psychiatric comorbidity. HS and psoriasis share the same pathogenic inflammatory pathway: upregulation of cytokines TNFa, IL-12/23 and IL-17. This implies that they could respond both to the same biologic treatments. The co-occurrence of these diseases in a pediatric setting is unknown [29].

We report a peculiar case of coexisting HS and psoriatic disease in a pediatric patient with chromosome 17q21.31 microduplication syndrome.

The C-reactive protein levels decreased during adalimumab therapy in patients with psoriasis who experienced suboptimal response to previous therapies. Clinical response was associated with greater CRP reductions overall and in substudies E and M, but not P. CRP reductions correlated.

## 2. Case Presentation

A 10-year-old female patient came to our department for flexural lesions that had risen up in about 6 months, and itchy patches on the head and neck region. She was slightly overweight (BMI = 20), and had a positive familial history for psoriatic arthritis (mother) but not for HS. The patient presented: low birthweight, microcephaly, facial dysmorphisms, lumbar hyperlordosis, walking difficulties, global psychomotor developmental delay and learning disabilities. A comparative genomic hybridization (CGH) array was carried, revealing a 2.5 Mb de novo microduplication in the 17q21.31 chromosomal region.

Dermatological examination put in evidence double-ended pseudocomedones, follicular nodules and pustules, painful subcutaneous abscesses and sinus tracts distributed in the genital area and inguinal folds, compatible with the diagnosis of HS (Hurley stage II-HS Physician Global Assessment PGA 3). Her International Hidradenitis Suppurativa Severity Score System (IHS4) was 11.

Furthermore, erythemato-squamous psoriatic patches on the scalp, on the retroauricolar folds, on the elbows, bilaterally, and on the lateral aspect of the right ankle, were present. The patient complained about ankle pain and stiffness: considering ultrasound findings (mild distension of tibiotalar joint capsules and peroneal proliferative tenosynovitis bilaterally) and positive family history, a diagnosis of psoriatic arthritis (PsA) was provided. Beside the increase of ESR (58 mm/h) and CRP (2.0 mg/dL), laboratory tests were within normal range. Thus, in order to treat simultaneously HS and psoriasis, it was decided to start a biological therapy with adalimumab at a dosage of 40 mg administered subcutaneously every two weeks. After 24 weeks of treatment, an important improvement of both HS and psoriasis scores was observed: the psoriasis area and severity index (PASI) score reduced from 15 to 3, Children’s dermatology life quality index (CDLQI) reduced from 25 to 10, pain visual analog scale (VAS) reduced from 8 to 4, HS-PGA reduced from 3 to 2, and IHS4 from 11 to 3.

Moreover, ultrasound and clinical examination confirmed PsA improvement.

## 3. Discussion

Chromosome 17q21.31 microduplication syndrome represents a rare genomic disorder that was described for the first time in 2007. Only 10 cases have been reported until now, with duplication ranging from 485 to 763 kb involving mostly genes CRHR1, MAPT, IMP5, STH and KANSL1. A broad clinical spectrum has been described in association with this syndrome: facial dysmorphism, microcephaly, short stature, abnormal digits, Achilles tendon retraction, global hirsutism, severe obesity, psychomotor retardation, behavioral problems with outbursts of temper, and poor social interaction reminiscent of autism spectrum disorder [30].

However, this genetic alteration has not been reported yet in association with acne, HS and psoriasis. Among the genes encoded by the 17q21.31 regions, corticotrophin-releasing hormone receptor-1 (CHRR-1) has been implicated in the pathogenesis of some inflammatory dermatoses [31].

In particular, the sebaceous gland exhibits an independent peripheral endocrine function and expresses receptors for neuropeptides, such as CRHR1. The capability of hypothalamic CRH to induce lipidsynthesis, induce steroidogenesis and interact with testosterone and growth hormones, implicates a possibility of its involvement in the clinical development of acne and HS. Additionally, weight gain and hyperandrogenism are common risk factors for the development of both these cutaneous pathologies [32].

Moreover, an aberrant cutaneous CRH/CRH-R1 system has been demonstrated in psoriatic lesions, which offers further evidence that CRH/CRH-R1 is actively involved in the pathogenesis of psoriasis; in particular, it has been postulated that the aberrant expression of CRH/CRH-R1 in the skin have a clear pro-inflammatory role (modulating the expression of several cytokines, such as Interferon-γ, Tumor Necrosis Factor-α,Interleukin-6 and 18) and leads to an abnormal differentiation and proliferation in keratinocytes [33]. In addition, an overexpression of CRHR1 in psoriatic lesions could contribute to increased PASI scores, as already reported in literature [34].

## 4. Conclusions

To our best knowledge, this is the first case described in literature of psoriatic disease and HS associated with chromosome 17q21.31 microduplication syndrome. We hypothesize that gene CRHR1 duplication included in the 17q21.31 chromosomal region could also have a role in the pathogenesis of the chronic inflammatory dermatological comorbidities.

## Data Availability

All data generated or analysed during this study are included in this. Further enquiries can be directed to the corresponding author.

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
