# Peer review of "A Case of Psoriatic Disease and Hidradenitis Suppurativa in a Child with Chromosome 17q21.31 Microduplication Syndrome"

_children, 2023, doi:10.3390/children10060931_

Round 1
Reviewer 1 Report
This is an interesting report but I have some queries and comments
· Introduction: Given the paper has a genetic background, I would specify that some patient with HS have underlying monogenic variation particularly in genes coding for proteins of the É£ secretase complex (PMID: 35401657)
· Line 51: what genomic test was used? If it is microaaray-based comparative genomic hybridization – do you have images to include?
· Patient characteristics: What was the patient’s BMI, did they have a family history of hidradenitis suppurativa? The patient was 10 years old… how old was she when she started experiencing symptoms?
· Would you be able to use a dynamic severity scoring system to rate the improvement in HS after adalimumab (example IHS4)
Reviewer 2 Report
Thank you for the opportunity to read this manuscript. It's interesting because the genetic factors in HS play a role.
This is case report. Presenting HS and psoriatic patient. Paper is interesting however, I have some concerns :
1) The Introduction lacks important information about the nature of the disease and its treatment i.e. no sentence about the reduced quality of life in these patients that completely disables normal functioning e.g. PMID : 35893421 There is a missing sentence about the treatment of HS itself : Treatment with antibiotics and new biologic drugs, small molecules drugs etc. Also missing is information that HS is treated surgically PMID : 36686007 and that there are recent methods of surgical treatment of HS like co-graft ADM and STSG , PMID : 36004913 and the very use of ADM in surgery and surgical treatment of HS e.g. PMID : 36359387
2) Other Genetic factors should be mentioned in HS
The cause of hair follicle plugging is still unclear and subject to scientific debate, although immune and genetic factors, particularly variants in genes encoding ɣ secretase complex proteins (NCSTN encoding nicastrin; PSEN1 for presenilin-1; PSENEN for presenilin enhancer; gamma-secretase subunit) doi : https://doi.org/10.3390/jcm12062112
3) You should definitely extend introduction section and reference section which is quite poor (11 items)
4) Reference section should be in MDPI style.
5) I am waiting for extend research on CRHR1 duplication. How many patients with this duplication have you found?
Im sure that authors will adress well all suggestions - this paper is important, but it needs some corrections.
Sincerely yours,
Reviewer 3 Report
This is a very interesting case report.
I have some minor comments:
1. In the Introduction information is very general. It would be more relevant to point out characteristics of perdiatric psoriasis and/or HS. Especially where treatment information is given. E.g. Is extensive surgery in very young pediatric patients a choice? tetracyclines do not have the same status as in adults. The same applies to comorbidities. Are the same comorbidities typical for children as for adults?
2. You mention that HS and psoriasis are often associated. Then you mention hpow this is a rare case. Contradiction. Perhaps rather mention that HS is rare in children. It may be useful also to mention the prevalence of HS in children and if available HS+psoriasis in children without this syndrome. This way you point out of your case is more special than other coocuring pediatric cases of HS and pso
3. Please mention ehether isntruments like PASI, DLQI, VAS, etc were modified for pediatric use. Did you use the children specific instruments?
4. Most references cite adult populations
Round 2
Reviewer 1 Report
Thank you for updating the manuscript.
It would have been great to have images in the manuscript.
Author Response
Response to Reviewer 1:
Unfortunately, we have been not able to provide graphic support to our work.
We added further informations in the introduction section (line 43-50 and 70-76) and three new references (3-5 and 23).
Thank you for your observations that allowed us to expand and improve our manuscript.
Reviewer 2 Report
Authors well adressed all suggestions.
i strongly recommend to accept this paper in current form
Author Response
Response to Reviewer 2:
Thank you.
We added some new informations in the introduction section (see line 43-50 and 70-76, and ref. 3-5 and 23)
We are very grateful for your precious observations thant allowed us to expand and improve our manuscript.